# Modulating PKCα Activity to Target Wnt/β-Catenin Signaling in Colon Cancer

**DOI:** 10.3390/cancers11050693

**Published:** 2019-05-18

**Authors:** Sébastien Dupasquier, Philippe Blache, Laurence Picque Lasorsa, Han Zhao, Jean-Daniel Abraham, Jody J. Haigh, Marc Ychou, Corinne Prévostel

**Affiliations:** 1UCL Louvain, IREC, 1348 Louvain-la-Neuve, Belgium; seb.dupasquier@laposte.net; 2Campus Val d’Aurelle—Research team “Integrative cancer research for personalized medicine in digestive oncology”, IRCM U1194, University of Montpellier, ICM, CNRS, CHU, 34298 Montpellier, France; philippe.blache@inserm.fr (P.B.); laurence.lasorsa@icm.unicancer.fr (L.P.L.); zhaohan1204@gmail.com (H.Z.); jean-daniel.abraham@inserm.fr (J.-D.A.); Marc.Ychou@icm.unicancer.fr (M.Y.); 3Research Institute in Oncology and Hematology, University of Manitoba, Winnipeg, MB R3E 0V9, Canada; Jody.Haigh@umanitoba.ca

**Keywords:** colorectal cancer, protein kinase Cα, Wnt/β-catenin, tumor suppressor, drug target

## Abstract

Inactivating mutations of the tumor suppressor Adenomatosis Polyposis Coli (APC), which are found in familial adenomatosis polyposis and in 80% of sporadic colorectal cancers (CRC), result in constitutive activation of the Wnt/β-catenin pathway and tumor development in the intestine. These mutations disconnect the Wnt/β-catenin pathway from its Wnt extracellular signal by inactivating the APC/GSK3-β/axin destruction complex of β-catenin. This results in sustained nuclear accumulation of β-catenin, followed by β-catenin-dependent co-transcriptional activation of Wnt/β-catenin target genes. Thus, mechanisms acting downstream of APC, such as those controlling β-catenin stability and/or co-transcriptional activity, are attractive targets for CRC treatment. Protein Kinase C-α (PKCα) phosphorylates the orphan receptor RORα that then inhibits β-catenin co-transcriptional activity. PKCα also phosphorylates β-catenin, leading to its degradation by the proteasome. Here, using both in vitro (DLD-1 cells) and in vivo (C57BL/6J mice) PKCα knock-in models, we investigated whether enhancing PKCα function could be beneficial in CRC treatment. We found that PKCα is infrequently mutated in CRC samples, and that inducing PKCα function is not deleterious for the normal intestinal epithelium. Conversely, di-terpene ester-induced PKCα activity triggers CRC cell death. Together, these data indicate that PKCα is a relevant drug target for CRC treatment.

## 1. Introduction

Variation in the activity of the Wnt/β-catenin signaling pathway along the intestinal epithelium crypt/surface axis is a key regulatory element of intestinal homeostasis (Figure 1). Indeed, the Wnt/β-catenin pathway is strongly activated within the crypt where it stimulates the proliferation of intestinal epithelial cells and ensures the constant renewal of the intestine epithelium. Moreover, the gradual decrease of Wnt/β-catenin activity from the bottom of the crypt to the surface of the intestinal epithelium is also critical for maintaining the physiological balance between proliferating and differentiating cells [1]. Wnt/β-catenin signaling activity is regulated through β-catenin—the transducer of this signaling pathway—by the APC/GSK3-β/axin destruction complex [2] (Figure 2). In the Wnt-off state, this complex phosphorylates β-catenin that is then ubiquitinated and degraded by the proteasome. In the Wnt-on state, the β-catenin destruction complex is recruited to the plasma membrane and is inactivated through a dishevelled-dependent mechanism [3,4]. Then, β-catenin is stabilized, accumulates in the nucleus, binds to T-Cell transcription Factor (TCF) and co-transcriptionally activates Wnt/β-catenin target genes (https://web.stanford.edu/group/nusselab/cgi-bin/wnt/target_genes). 

In inherited familial adenomatosis polyposis—the major hereditary disease leading to colorectal cancer (CRC)—and in up to 80% of sporadic CRC, genetic alterations of *APC* (Adenomatous Polyposis Coli) result in constitutive activation of Wnt/β-catenin signaling. These alterations include loss of heterozygosity (in about 30–40% of CRC) and point mutations that lead to the expression of truncated versions of the APC protein without the C-terminal region, a critical region for β-catenin destruction complex scaffolding [5] (Figure 2). In these pathological situations, β-catenin is constitutively stabilized in intestinal epithelial cells, and induces the sustained expression of Wnt/β-catenin target genes, including *MYC* that is considered the master gene responsible for the oncogenic activity of this pathway [6]. 

No therapy targeting the activity of the Wnt/β-catenin pathway is currently used to treat CRC, partly due to the lack of targets downstream of APC. However, several observations indicate that Protein Kinase C-α (PKCα) might fulfill this criterion. PKCα is a member of the Protein Kinase C (PKC) family that includes ten serine/threonine kinases encoded by nine distinct genes. The first members were isolated 41 years ago from mammalian brain extracts [7] and rapidly became the subject of intensive research in the cancer field due to their ability to be directly activated by phorbol ester tumor promoters [8]. However, despite more than 6000 citations linking PKC to cancer in PubMed (https://www.ncbi.nlm.nih.gov/pubmed), PKC role as oncogenes or tumor suppressors has been the subject of discussion for a long time. This could be one of the main reasons why PKC is not considered to be a prime target for cancer therapy. Significant evidence indicates that PKCα acts as a tumor suppressor in intestinal epithelial cells: (i) PKCα inactivating mutations have been detected in CRC [9]; (ii) PKCα expression is frequently decreased in human CRC and its down-regulation induces intestinal tumorigenesis in mouse models [10,11]; (iii), decreased PKCα expression promotes CRC cell growth and tumorigenicity [12]; (iv) *PRKCA* (the gene encoding mouse PKCα) knock-out in mice is associated with the development of spontaneous intestinal polyps, and with higher tumor aggressiveness and reduced survival in the APCMin/+ background [11]; and (v) PKCα activity significantly reduces CRC cell invasion and growth [13]. Therefore, PKCα can exert its tumor suppressor activity in the intestinal epithelium by antagonizing the activity of the Wnt/β-catenin signaling pathway. Indeed, PKCα activity is inversely correlated with that of the Wnt/β-catenin pathway: it gradually increases from the cryptic compartment to the surface of normal intestinal epithelium, while it is significantly reduced in CRC in which the Wnt/β-catenin signaling pathway is constitutively activated (Figure 1) [10,14,15,16,17]. Two studies clearly described two distinct mechanisms by which PKCα can inhibit the Wnt/β-catenin pathway activity by targeting the β-catenin transducer, downstream of APC (Figure 2). Lee et al. (2010) [18] reported that PKCα-induced phosphorylation of the orphan receptor RORα results in β-catenin co-transcriptional activity inhibition and in the trans-repression of Wnt/β-catenin target genes. Gwak et al. (2006) [19] demonstrated that PKCα-induced phosphorylation of β-catenin promotes β-catenin ubiquitination and degradation by the proteasome. 

An additional attractiveness of PKC as a therapeutic target is the availability of many pharmacological compounds that can modulate its activity, many of which have been used in clinical trials to evaluate their potential anti-tumor effects (https://clinicaltrials.gov/). However, despite the findings suggesting that PKCα is a potential drug target candidate for inhibiting the constitutively activated Wnt/β-catenin pathway in CRC, it is not known whether increasing PKCα activity is beneficial for fighting CRC. Using both in vitro and in vivo PKCα knock-in models, we provide evidence that PKCα is a relevant drug target to be stimulated in CRC. 

## 2. Results

### 2.1. Infrequent Inactivating Mutations of PKCα in Human Intestinal Tumors

From the COSMIC database, we listed 21 distinct PKCα non-silent mutations in intestinal tumors (http://cancer.sanger.ac.uk/cosmic) (Figure 3 and list in Appendix A). Most of them are localized in two critical domains for PKCα activity: the C1 domain that is required for binding of diacylglycerol (DAG)/phorbol ester activators, and the kinase domain that is responsible for PKCα phosphorylation activity. In the kinase domain, the A444V (also found in endometrial and breast cancers) and D481E mutations have been characterized as LOF mutations [9]. This indicates that mutations within this domain can decrease or potentially abolish PKCα activity in CRC cells. However, the frequency of PKCα mutations in CRC is low (5.35% of CRC in non-Western patients and only 2.78% of CRC in Western patients) (COCA-CN and COAD-US projects, http://dcc.icgc.org). In agreement, sequencing analysis of the eight human CRC cell lines used in this study showed that HCT116, RKO and HT29 cells carried silent PKCα mutations, whereas only DLD1 cells exhibited a non-silent mutation within the PKCα coding sequence, resulting in a non-previously described methionine to isoleucine substitution at position 299 of the protein (Appendix A). Given this low frequency and the heterozygous status of PKCα mutations in CRC (Appendix A), it can be considered that PKCα integrity is preserved in most human CRC, a critical criterion for inducing PKCα activity in CRC cells. 

### 2.2. Increasing PKCα Activity in the Healthy Intestinal Epithelium Is Not Deleterious

As PKCα exhibits tumor suppressor properties in the intestinal epithelium, decreasing PKCα expression should promote tumor development or tumor aggressiveness, while increasing PKCα expression should protect from cancer development. Consistent with this hypothesis, *PRKCA* knock-out has been associated with higher tendency to develop spontaneous intestinal tumors in wild type mice, and with more aggressive tumors and reduced survival rate in APCmin/+ mice [11]. However, PKCα is also involved in cell differentiation [20] and trans-epithelial resistance [21]. Therefore, we first assessed the phenotypic consequences of increased PKCα function in healthy intestinal epithelium. To this aim, we overexpressed PKCα in the intestinal epithelium by generating conditional homozygous *PRKCA* cDNA ROSA26 knock-in transgenic mice (hereafter referred to as R26-PKCα^Tg/Tg^ [22]) that were intercrossed with villin-Cre mice that express the Cre recombinase in intestinal epithelial cells of villi and crypts [23] (see Appendix A and Appendix A for technical information). Western blot analysis (Figure 4A) and immunostaining of intestinal sections (Figure 4B) showed that PKCα was overexpressed in the intestinal epithelium of villin-Cre;R26-PKCα^Tg/Tg^ mice compared with R26-PKCα^Tg/Tg^ littermates. PKCα immunostaining was particularly increased at the plasma membrane of differentiated and functionally mature cells in both the small intestine villi and the colon epithelium surface (white arrowheads in magnifications of Figure 4B). As PKCα binding to the plasma membrane reflects its activation, this observation suggests that the increase of PKCα expression correlated with an increase of activity in the intestinal epithelium of villin-Cre;R26-PKCα^Tg/Tg^ mice. Nevertheless, villin-Cre;R26-PKCα^Tg/Tg^ males and females developed normally and weighed the same as R26-PKCα^Tg/Tg^ littermates (Figure 4C). Histological analysis of whole intestinal tissue sections of 31-week-old mice revealed that the number and size of villi in the small intestine and the number of crypts in small intestine and colon were significantly increased in villin-Cre;R26-PKCα^Tg/Tg^ mice compared with control littermates (Figure 4D) suggesting that the increase of PKCα activity may possibly interfere with the intestinal epithelium morphogenesis. However, we did not detect any other abnormality, including in older mice (up to 76 weeks of age), indicating that increasing PKCα expression/activity in the healthy intestinal epithelium is not deleterious. 

### 2.3. Increasing PKCα Activity in DLD-1 CRC Cells Disrupts Cell Morphology and Causes Cell Cycle Arrest

In CRC cells, PKCα inhibits Wnt/β-catenin signaling activity downstream of APC [18,19]. To determine whether stimulating PKCα function in CRC cells with mutated APC inhibited cell growth, we generated a DLD-1 CRC cell line in which PKCα overexpression is induced by incubation with doxycycline (DLD-1-PKCα cell line). PKCα was weakly detected in parental DLD-1 cells (Figure 5A, upper panel) and in DLD-1-PKCα cells which were not exposed to doxycycline (Figure 5A, lower panel), in agreement with the low endogenous PKCα levels in CRC cells. Upon exposure to doxycycline for 24 h, exogenous PKCα expression increased in DLD-1-PKCα cells and accumulated in the cytoplasm, but not at the plasma membrane (Figure 5A, lower panel). This observation suggests that overexpressed PKCα is not activated, and that increasing PKCα expression is not sufficient to induce PKCα function in DLD-1 CRC cells. Incubation with 1 nM of phorbol 12-myristate 13-acetate (PMA; the prototypic phorbol ester) [8] did not have significant effects in both DLD-1 and DLD-1-PKCα cells, which is in agreement with the low endogenous PKC levels in CRC cells. Conversely, incubation of DLD-1-PKCα cells with 1 μg/mL doxycycline and 1 nM PMA for 24 h induced the translocation of overexpressed PKCα to the plasma membrane (Figure 5A, lower panel), formation of membrane protrusions (empty arrowhead) and disruption of DLD-1 cell morphology with large buds (solid arrowheads). These changes were associated with a drastic remodeling of the actin network (Figure 5A, lower panels, phalloidin staining), relocation of actin, E-cadherin and β-catenin into membrane protrusions and large buds (Figure 5A, lower left panels, black and solid arrowheads), decrease of the cell size (Figure 5B), and cell cycle arrest in the G0/G1 phase (Figure 5C), as previously reported in colon cancer cell lines including DLD-1 cells [12,13,24,25,26,27]. This phenotype correlated with inhibition of the activity of the Wnt/β-catenin pathway, as indicated by the decrease of TOPflash luciferase reporter activity measured in DLD-1-PKCα cells incubated with both PMA and doxycycline compared with untreated cells (Appendix A). We obtained similar results upon overexpression of stabilized S^33^β-catenin, a dominant-positive mutant of β-catenin where serine 33 is mutated into tyrosine that prevents β-catenin degradation [28]. In addition, western blot experiments (Appendix A) evidenced a decrease in active β-catenin. Since activating PKCα function and inhibiting the Wnt/β-catenin signaling both cause cell cycle arrest in the G1 phase (Figure 4C and [29]), we conclude that PKCα-induced colon cancer cells growth arrest is mediated through inhibition of the Wnt/β-catenin signaling. However, we cannot exclude the involvement of additional mechanisms.

### 2.4. Increasing PKCα Activity in DLD-1 CRC Cells Prevents Cell Growth and Promotes Cell Death

To evaluate the effects of sustained PKCα activity on CRC cell growth capacity, we cultured parental DLD-1 and DLD-1-PKCα cells at very low concentration (1000 cells/10 cm dish) in the presence or absence of 1 μg/mL doxycycline and/or 1 nM PMA. After 10 days, we stained cultures with crystal violet (Figure 6A), and quantified the growth capacity in the different conditions by calculating the variation of the total area occupied by the clones (Figure 6B) and of the Optical Density at 570 nm (OD_570 nm_) relative to untreated cells (control) (Figure 6C). The results clearly demonstrated that (doxycycline + PMA)-induced PKCα expression and activity in DLD-1-PKCα cells inhibited cell growth by more than 80%. Moreover, quantification of dead cells after trypan blue staining indicated that (doxycycline + PMA)-induced PKCα expression/activity promoted cell death (about 75% of dead cells compared with 10–30% in controls) (Figure 6D). Incubation of DLD-1-PKCα cells with 1 μg/mL doxycycline and increasing concentrations of PMA for 24 h showed that the extent of cell death was PMA dose-dependent (phase contrast images in Figure 6E, lower panels). 

### 2.5. Di-Terpene Esters Inhibit CRC Cell Growth with Different Potencies

The previous results demonstrated that doxycycline-induced PKCα overexpression inhibits CRC cell growth and promotes their death upon PMA stimulation. Although PKCα is infrequently mutated in CRC cells, it has been reported that PKCα expression is decreased in CRC [30], raising the question of whether PKCα overexpression and drug-induced activation are sufficient to induce its tumor suppressor function. Moreover, the isozyme specificity of PKC activators is questionable. Indeed, PKC activators interact with the DAG binding site that is located in the highly conserved C1 domain of classical (α, βI, βII, γ) and novel (δ, ε, θ, η) PKC isoforms. Consequently, none of the available PKC activators is highly selective for a specific isoform. The di-terpene ester PEP005 shows preferential selectivity for PKCα and δ and induces growth arrest of some CRC cell lines, such as HT29 and Colo205 cells [31,32]. In agreement, crystal violet staining of DLD-1-PKCα cells incubated with increasing concentrations of PMA or PEP005 in the presence or not of doxycycline for 10 days clearly demonstrated that like PMA, PEP005 inhibited DLD-1 cell growth in a concentration- and a PKCα-dependent manner, although the required concentrations were 10 times higher than those for PMA (Figure 7A). In control conditions (without doxycycline), PMA inhibited the growth of DLD-1-PKCα cells from 10 nM and PEP005 from 100 nM. Conversely, in doxycycline-treated cells to induce PKCα overexpression, cell growth was inhibited by both compounds at ten times lower concentrations (Figure 7A). Moreover, in a panel of CRC cell lines, PMA and PEP005 inhibited cell growth with different potencies. PMA strongly inhibited CRC cell growth from 1 nM, but only in a limited number of cell lines (HT29, SW480, SW620) (Figure 7B). All PMA-sensitive CRC cell lines (HT29, SW480, SW620) displayed a Microsatellite Stable (MSS) status while the others (DLD-1, HCT116, RKO and CT26) exhibited a Microsatellite Instable (MSI) status. Conversely, PEP005 did not have any effect at 1 nM whatever the cell line (Figure 7B), but was significantly more efficient than PMA at 100 nM and in all tested CRC cell lines (Figure 7C). These findings suggest that di-terpene esters are natural compounds with clinical interest for CRC treatment.

## 3. Discussion

Despite significant prevention and screening measures implemented in recent years, CRC remains the third leading cause of cancer-related deaths in industrialized countries [33].

The constitutively activated Wnt/β-catenin signaling pathway is the prime target in CRC; however, no therapy to antagonize this pathway is currently available in the clinic, despite extensive research in this field [34]. Zhang et al. (2016) recently reported promising results with the small-molecule Truncated APC Selective Inhibitor-1 (TASIN-1) [35]. TASIN-1 selectively targets APC truncated mutants in CRC cells (Figure 2) and specifically kills cancer cells with APC truncations in xenografts models and in a genetically engineered CRC mouse model with minimal toxicity. In the present study, we provide evidence that PKCα, a physiological enhancer of β-catenin degradation [19] and a repressor of β-catenin co-transcriptional activity ([18] and see Appendix A), also could be an attractive candidate drug target for CRC treatment.

### 3.1. PKCα Integrity Is Preserved in Most Human CRCs

*PRKCA* mutations can be found in many human tumors, sometimes with high frequencies (e.g., 90.71% in skin cancer and 88.00% in melanoma) (MELA-AU, SKCA-BR, http://dcc.icgc.org). However, these observations must be tempered by the results displayed in the TCGA database (TCGA-SKCM, https://portal.gdc.cancer.gov) that identify *PRKCA* mutations in only 4.48% of skin cancers. This discrepancy can be explained by the use of distinct sequencing strategies i.e., Whole Genome Sequencing (WGS) for ICGC and Whole Exome Sequencing (WXS) for TGCA. Besides, in a recent study, Rosenberg et al. identified a recurrent point mutation in *PRKCA* as a hallmark of chordoid gliomas [36]. Conversely, *PRKCA* mutations are infrequent and heterozygous in CRC. This indicates that PKCα integrity is preserved and that PKCα activity could be stimulated in CRC cells. 

### 3.2. Inducing PKCα Function Is Not Deleterious for the Normal Intestinal Epithelium

*PRKCA* knock-out is sufficient to induce tumor formation in the intestine of wild-type mice and to promote tumor progression in the intestine of the APCmin/+ mice, indicating that PKCα plays a significant role in maintaining the intestinal epithelium homeostasis [11]. Conversely, to our knowledge, *PRKCA* cDNA was knocked-in only in the epidermis of transgenic mice where PKCα overexpression did not have any significant effect, except upon PMA induction of its activity that resulted in a striking inflammatory response, without tumor promotion [37]. Here, we found that targeted *PRKCA* cDNA knock-in in intestinal epithelial cells resulted in a significant increase of PKCα expression at the plasma membrane of differentiated and functionally mature cells in both the small intestine villi and the colon epithelium surface. This indicates that differently from what was reported for the epidermis, PKCα overexpression in the intestinal epithelium correlates with an increase of its activity. The increase of PKCα function in the intestine did not lead to deleterious phenotypic changes, detectable alterations of the intestinal function, and/or effects on food intake. 

### 3.3. Inducing PKCα Activity Potently Inhibits the Growth and Triggers the Death of CRC Cells 

PKCα overexpression is not sufficient to increase PKCα activity in CRC cells, as indicated by its expression mainly in the cytoplasm and not at the plasma membrane (Figure 5A, lower panel). Only after incubation with the PKCα activators PMA and PEP005, including at concentrations as low as 1 nM for PMA and 10 nM for PEP005, PKCα-overexpressing DLD-1 cells showed morphological changes, stopped growing, and died, clearly demonstrating the anti-tumor effect of PKCα activity in CRC cells.

### 3.4. PKCα Activity Is Drug-Targetable

Very soon after their discovery, PKC members were identified as binding receptors for phorbol ester tumor promoters, such as PMA [8], suggesting that PKC members are oncogenes. Consequently, many efforts were made to develop PKC inhibitors with extremely disappointing results in clinical trials. Twenty years ago, we [33,38] and more recently, Newton and co-workers [9] identified tumor-associated LOF mutations in PKC members, including PKCα, suggesting a tumor suppressive role. In addition, Black and co-workers clearly demonstrated the tumor suppressor role of PKCα in the intestine [39,40]. These observations suggest that previous therapeutic strategies targeting PKC could have failed because they were mainly designed to inhibit PKC activity, instead of stimulating it [41]. Indeed, very few PKC activators have been tested in clinical trials in cancer [42]. PKC activators are natural compounds, such as the macrolide lactone bryostatin extracted from marine bryozoan species, and the di-terpene esters PMA and PEP005, both extracted from euphorbia plants (Table 1). Bryostatin is slightly more selective for PKCε and has been extensively tested in different cancers, including CRC, but without benefit, possibly due to the oncogenic activity of PKCε in colon cells [43]. Di-terpene esters exhibit a good affinity for PKCα and seem more promising. Indeed, systemic administration of PMA induced temporary remission with only mild toxicity in patients with myeloid leukemia, most of whom were refractory to conventional therapies [44,45]. Fang et al. [46] reported that combining PMA with the tyrosine kinase inhibitor imatinib might overcome a poor response to conventional doses of imatinib in patients with chronic myelogenous leukemia. However, in mice, PMA has been associated with tumor promotion upon repeated applications on the skin [47]. More promisingly, the di-terpene ester PEP005 (ingenol-3-angelate) isolated from the sap of *Euphorbia peplus*, a traditional home remedy for warts, corns and even skin cancer [48], treats effectively skin lesions and has been tested in phase III clinical trials by Peplin Ldt. Unlike PMA, PEP005 does not seem to have a tumor-promoting activity upon repeated application on normal mouse skin [49]. Here, we found that PEP005 was poorly effective at 1 nM, but efficiently inhibited the growth of all tested CRC cells when used at 100 times higher concentrations. Together, these findings suggest that PEP005 and its derivatives should be considered to be promising therapeutic compounds for CRC treatment. PKCα expression is significantly decreased in colonic cancer cells. This could be a counter-argument to the relevance of targeting PKCα in CRC. However, PKCα can be detected in both CRC cell lines [50] and in human adenomas [17]. Interestingly, Assert et al. [17] even reported a significant increase of PKCα in the cytosolic fraction of colonic adenomas compared to normal neighboring mucosa, thus indicating, again that PKCα is still expressed but not activated in CRC cells. This observation further raises the question of whether the levels of PKCα detected in colon adenomas are evolving or not with tumor progression. Addressing this issue could be critical to clarify whether targeting PKCα would be more relevant at early, late, or any stages of the disease. Finally, focusing research on the identification of new and more selective PKCα activators and on the development of vectorization technologies to bring PKC activators to the tumor site should improve drug efficacy and reduce side effects [51]. 

## 4. Material and Methods

### 4.1. Culture of Colon Cancer Cells and Generation of the DLD-1-PKCα Cell Line

Human DLD-1, HCT116, RKO, SW480, SW620, HT-29 and mouse CT26 colon cancer cells from American Type Culture Collection (ATCC, Manassas, VA, USA) were cultured in DMEM medium (Gibco, Fisher Scientific, Illkirch, France) supplemented with 10% fetal bovine serum (FBS) (Eurobio, Les Ulis, France) and 1× penicillin/streptomycin (Gibco).

DLD-1 cells in which PKCα overexpression was induced with doxycycline were established by lentiviral infection with a vector carrying the pTRIPZ inducible system (Dharmacon, Brebieres, France) and the cDNA of interest instead of the turbo red fluorescent protein. Puromycin-resistant clones were selected and maintained in DMEM medium containing 10% FBS and 10 µg/mL puromycin (Invivogen, Toulouse, France). All experiments were conducted in DMEM medium containing 10% FBS and 10 µg/mL puromycin without (control) or with 1 μg/mL doxycycline (Sigma-Aldrich, Saint-Quentin Fallavier, France). PMA (Sigma-Aldrich) or PEP005 (Tocris, Bio-Techne Ltd., Lille, France) were added at the concentrations and for the times indicated in the legends to figures.

### 4.2. Characterization of the PKCα-Induced Colon Cancer Cells Growth Arrest

Cell growth was evaluated after 11 days of cell culture by image analysis (ImageJ Software) of scanned clones grown from 1000 cells/10 cm dish and stained by crystal violet or by OD_570nm_ measurements (Polarstar BMG Labtech, Champigny sur Marne, France) of cells seeded in 96-well plates (50 cells/well), stained with crystal violet (0.2% crystal violet; 2% ethanol) and lysed in 1% SDS. Cell number, volume and diameter were measured using the Scepter^TM^ Automated Cell Counter (Millipore, Saint Germain en Laye, France). Cell viability was quantified by counting living and trypan blue-stained dead cells using a Countess II automated cell counter (Thermo Fisher Scientific, Villebon sur Yvette, France), according to the manufacturer’s instructions.

Cell cycle was analyzed using a Propidium Iodide (PI) (Sigma-Aldrich) flow cytometry assay. Briefly, 2 × 10^6^ DLD-1-PKCα cells grown with/without (doxycycline + PMA) for 24 h, were washed with 1× PBS, trypsinized and centrifuged at 2000 rpm at 4 °C for 5 min. The resulting pellets were resuspended in 500 µL 1× PBS and fixed on ice in 1.5 mL of cold absolute ethanol for 10 min. Fixed cells were centrifuged at 2000 rpm at 4 °C for 10 min, washed in 1× PBS, and centrifuged at 2000 rpm at 4 °C for 10 min. They were then stained in 500 µL 1× PBS containing 40 µg/µL PI and 100 µg/µL RNase A at room temperature for 20 min. Cells were analyzed with a Gallios flow cytometer (Beckman Coulter, Krefeld, Deutschland).

### 4.3. Antibodies and Dyes

The rabbit anti-PKCα and mouse anti-β-actin (AC-15) antibodies and phalloidin were from Sigma-Aldrich. The mouse anti-E-cadherin and mouse anti-β-catenin antibodies were from BD Transduction Laboratories (Le Pont de Claix, France). The mouse anti-active β-catenin was from Millipore. The mouse anti-α-tubulin was from Santa-Cruz (Heidelberg, Germany). The Alexa Fluor 488-conjugated F(ab’)2 fragment of goat anti-rabbit IgG and Alexa Fluor 543-conjugated F(ab’)2 fragment of goat anti-mouse IgG and Hoechst were from ThermoFisher Scientific. HRP-conjugated anti-mouse and anti-rabbit antibodies were from Millipore.

### 4.4. Generation of the R26-PKCα^Tg/Tg^ and Villin-Cre;R26-PKCα^Tg/Tg^ Mice 

The knock-in construct was made by using the GateWay-compatible vector technology (Invitrogen, Villebon sur Yvette, France) to insert the human *PRKCA* cDNA between a floxed Neo/Stop cassette and a pIRES-EGFP sequence in the pRosa26-DEST vector [22]. After insertion, the targeting cassette was entirely sequenced and the vector was linearized by *Xho*I digestion before electroporation in 129/Sv ES cells. G418-resistant ES clones containing the recombinant allele (PRKCA-KI) within the Rosa26 locus were identified by PCR (Appendix A) using the primers listed in Appendix A. Two positive ES clones were injected in C57BL/6J blastocysts to generate chimeric mice. After breeding with C57BL/6J mice, germline transmission was checked by PCR analysis using the pIRES/GFP-GFPrev and the P3–P5 primer pairs to discriminate between recombinant and wild type alleles (Appendix A and Appendix A). F1 germline heterozygous mice (R26-PKCα^Tg/wt^) were then inter-crossed to produce F2 homozygous R26-PKCα^Tg/Tg^ mice. The established R26-PKCα^Tg/Tg^ line was finally bred with villin-Cre mice [23] to specifically target expression of the *PKCA* knocked-in gene in the intestinal epithelium upon excision of the Neo/Stop cassette. The resulting heterozygous villin-Cre; R26-PKCα^Tg/wt^ mice were backcrossed with R26-PKCα^Tg/Tg^ mice to obtain homozygous villin-Cre; R26-PKCα^Tg/Tg^ mice.

### 4.5. Characterization of the Mouse Phenotypes

Four males and four females R26-PKCα^Tg/Tg^ and villin-Cre;R26-PKCα^Tg/Tg^ littermates were monitored and weighted weekly from weaning to 34 weeks post-partum. Whole intestines of 14- and up to 76-week-old mice were collected, fixed as rolls, and embedded in paraffin. Then, 3 μm thick tissue sections were stained with hematoxilin and eosin (H&E) and processed for immunofluorescence analysis of PKCα expression. H&E-stained tissue sections were scanned (×20 magnification) using the Nanozoomer technology (Hamamatsu Photonics France, Massy, France). The number of villi and crypts in 2 μm was counted in each intestinal section (*n* = 4 areas per section) and their size (*n* = 10 per section) was calculated using the NDP.view2 software (version U12388-1, Hamamatsu city, Japan). Fresh intestinal epithelium was also collected by scraping the mice inner intestinal layer and frozen for western blot analysis. 

### 4.6. Hematoxylin and Eosin Staining, Immunohistochemistry 

H&E staining and immunohistochemistry were performed on tissue sections deparaffinized in xylene and gradually rehydrated in 100%, 96%, 70%, 30% alcohol. For H&E staining, tissue sections were incubated in 0.1% Mayer hematoxylin, decolorized in 70% and 95% alcohol, counterstained in 0.5% eosin, dehydrated in 100% alcohol, and mounted in Mountex (Sigma-Aldrich). For immunohistochemistry, tissue sections were unmasked in 10 mM citrate buffer pH6 at 99 °C for 30 min, and permeabilized in 0.2% Triton X-100. For immunocytochemistry, 50,000 cells were seeded on 12 mm round coverslips, cultured for 48 h, fixed in 4% paraformaldehyde and permeabilized in 0.2% Triton X-100. Unspecific binding was blocked by incubation in 1× PBS/0.1% Triton X-100/10% BSA. Cells were incubated with the primary antibodies at 4 °C overnight, and with Alexa Fluor-conjugated secondary antibodies at room temperature for 1 h. Cells were mounted in CityFluor^TM^ AF1 Plus DAPI mounting medium (Biovalley, Nanterre, France) and observed with a Leica DM6000 microscope (Objective ×40).

### 4.7. Western Blotting

Equal amounts of total protein extracts from mouse intestinal epithelium samples (50 μg) were separated on 10% acrylamide/Bis-acrylamide gels and then transferred onto PVDF membranes that were then blocked in 10% milk/1× PBS. Primary antibodies were added at 4 °C overnight, followed by HRP-conjugated secondary antibodies at room temperature for 1 h. Immunoblots were revealed using ECL western blotting detection reagents (Perkin Elmer, Villebon sur Yvette, France). 

### 4.8. Statistics

Statistical analyses were performed using the PRISM version 5.0, GraphPad software (San Diego, CA, USA). Data are the mean ± SEM (Student’s *t*-test). 

### 4.9. Ethics Approval

Animals housing and experiments were performed in accordance with the guidelines set forth by the Institute for animal research (Approval certificate D34-172-16). Protocols were approved by the institute ethics committee.

## 5. Conclusions

The constitutively activated Wnt/β-catenin signaling pathway is the prime target in CRC but no therapy to antagonize this pathway is currently available in the clinic. Considering that PKCα has been shown to be both a physiological enhancer of β-catenin degradation [19] and a repressor of β-catenin co-transcriptional activity [18], our study put PKCα back in the spotlight by demonstrating key criteria of a relevant target to be stimulated for treating CRC: (i) PKCα integrity is preserved in most human CRCs and can thus be functionally activated; (ii) inducing PKCα function in the intestine epithelium is not deleterious; (iii) inducing PKCα function triggers the death of CRC cells; (iv) PKCα activity is drug-inducible. In-depth studies on drug selectivity, vectorization and combination with other anti-cancer strategies should promisingly improve the therapeutic potential of activating PKCα in CRC in the future.

## Figures and Tables

**Figure 1 cancers-11-00693-f001:**
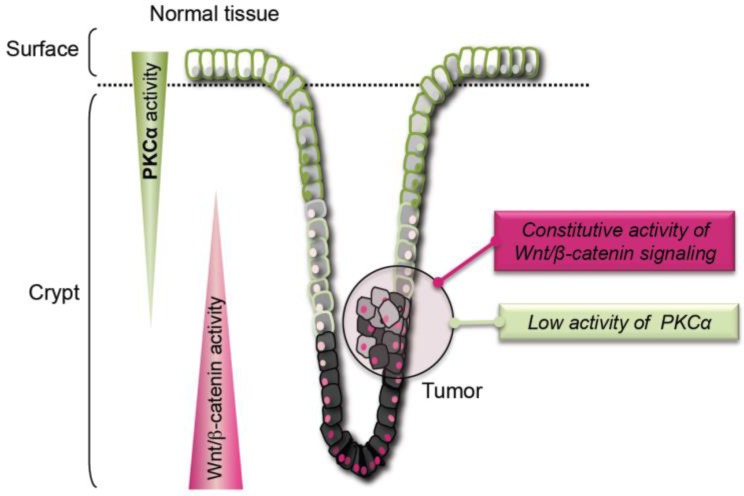
Schematic representation of Protein Kinase C-α (PKCα) and Wnt/β-catenin signaling activities along the crypt/surface axis of normal colon epithelium and in colon tumors. Note the inverse correlation between these activities in both normal and tumor tissues. In normal colon epithelium, Wnt-β-catenin activity is high in cryptic cells to maintain cell renewal and proliferation. It then decreases gradually along the crypt/surface axis with cell differentiation. Conversely, PKCα activity is low in proliferating cryptic cells and gradually increases upon differentiation. In tumor cells, Wnt/β-catenin activity is constitutively high and PKCα activity is low. The different intensity of pink in cell nuclei illustrates β-catenin nuclear accumulation level that is directly correlated with Wnt/β-catenin signaling activity. The intensity of the green color at the plasma membrane illustrates the amounts of PKCα targeted to the plasma membrane, an indication of PKCα activation.

**Figure 2 cancers-11-00693-f002:**
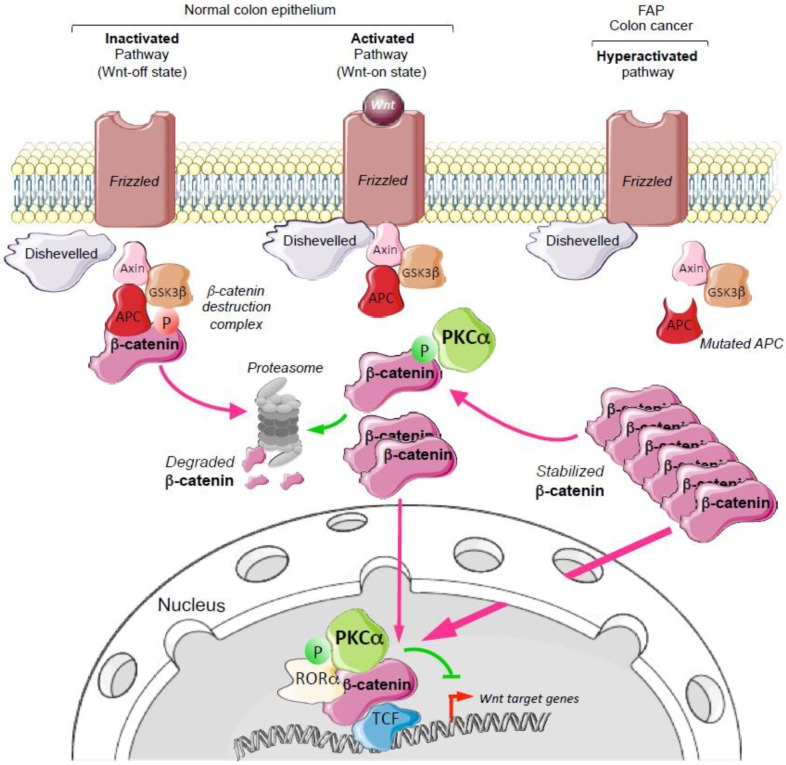
Schematic view of PKCα inhibitory activity on the Wnt/β-catenin signaling pathway in the normal colon epithelium and in colon cancer. In normal tissue, when the pathway is inactive (Wnt-off state, absence of Wnt ligand), β-catenin is phosphorylated by the kinase GSK3-β of the destruction complex. This is the signal for β-catenin ubiquitination and subsequent degradation by the proteasome. When the pathway is active (Wnt-on state, presence of Wnt ligand), the β-catenin destruction complex is recruited and inactivated at the plasma membrane through a dishevelled-dependent process. In this situation, β-catenin is neither phosphorylated nor ubiquitinated, and thus is not degraded by the proteasome; consequently, it accumulates in the cell including in the nucleus, where it activates T-Cell transcription Factor (TCF) to induce the expression of Wnt target genes. In colon cancer, the β-catenin destruction complex is constitutively inactive due to APC mutations. In this situation, β-catenin constitutively accumulates and co-transcriptionally activates the expression of Wnt target genes. PKCα, which acts downstream of APC, can inhibit the activity of the pathway in both normal and tumor cells: PKCα-induced phosphorylation of β-catenin targets β-catenin for degradation, while PKCα-induced phosphorylation of the orphan receptor RORα inactivates β-catenin co-transcriptional activity. FAP, Familial Adenomatous Polyposis.

**Figure 3 cancers-11-00693-f003:**
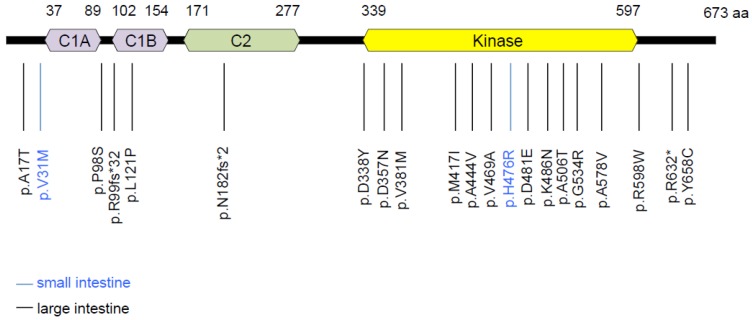
Non-silent PKCα mutations identified in both small intestine and colon tumors. (http://cancer.sanger.ac.uk/cosmic). p: protein sequence as the reference sequence; fs: shift of the reading frame. *: insertion of a stop codon.

**Figure 4 cancers-11-00693-f004:**
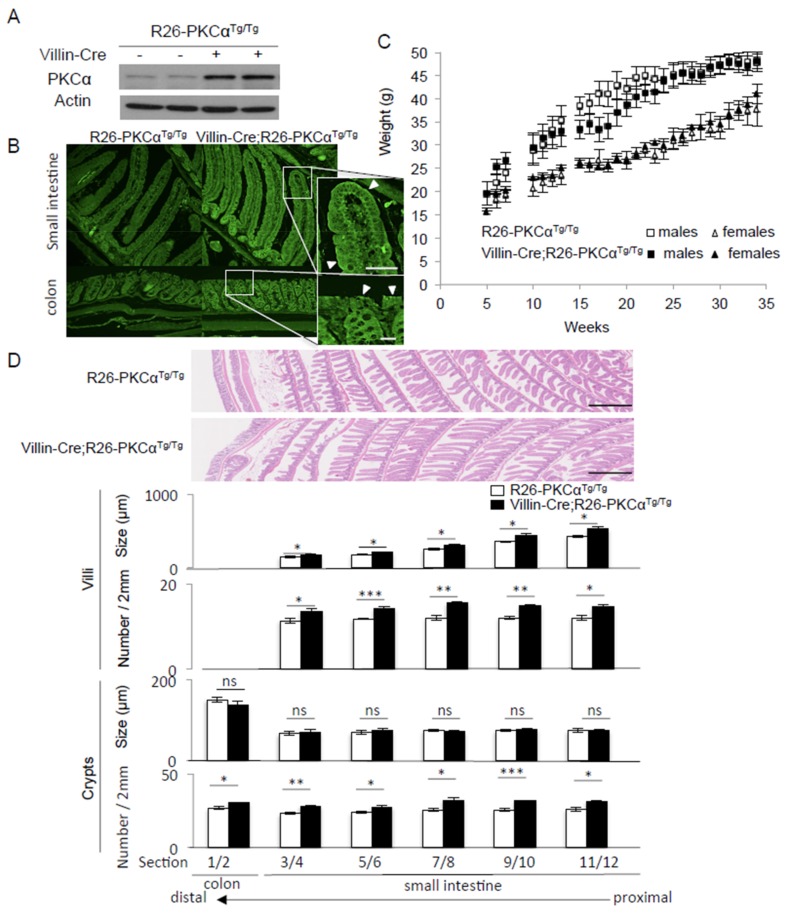
Targeted overexpression of PKCα in the mouse intestinal epithelium does not affect normal development and correlates with an increase in the size and number of villi and in the number of crypts. (**A**) Western blot analysis of intestinal epithelium samples, and (**B**) PKCα immunostaining of fixed whole intestine samples show the increase of PKCα expression and activity (i.e., increase of PKCα staining at the plasma membrane (white arrowheads in magnifications)) in the intestinal epithelium of villin-Cre;R26-PKCα^Tg/Tg^ compared with R26-PKCα^Tg/Tg^ 31-week-old mice (controls). Scale bar: 60 μm. (**C**) Weight curves of villin-Cre;R26-PKCα^Tg/Tg^ males (solid squares) and females (solid triangles) and R26-PKCα^Tg/Tg^ males (empty squares) and females (empty triangles) (**D**) Hematoxylin and eosin staining of fixed intestine showing an increase of the size (only villi) and the number of villi and crypts in villin-Cre;R26-PKCα^Tg/Tg^ compared with R26-PKCα^Tg/Tg^ mice. The lower panels show the quantification of the number and size of villi and crypts in 2 μm intestine tissue sections from the proximal small intestine to the distal colon of villin-Cre;R26-PKCα^Tg/Tg^ and R26-PKCα^Tg/Tg^ 31-week-old mice. Scale bar: 500 μm.

**Figure 5 cancers-11-00693-f005:**
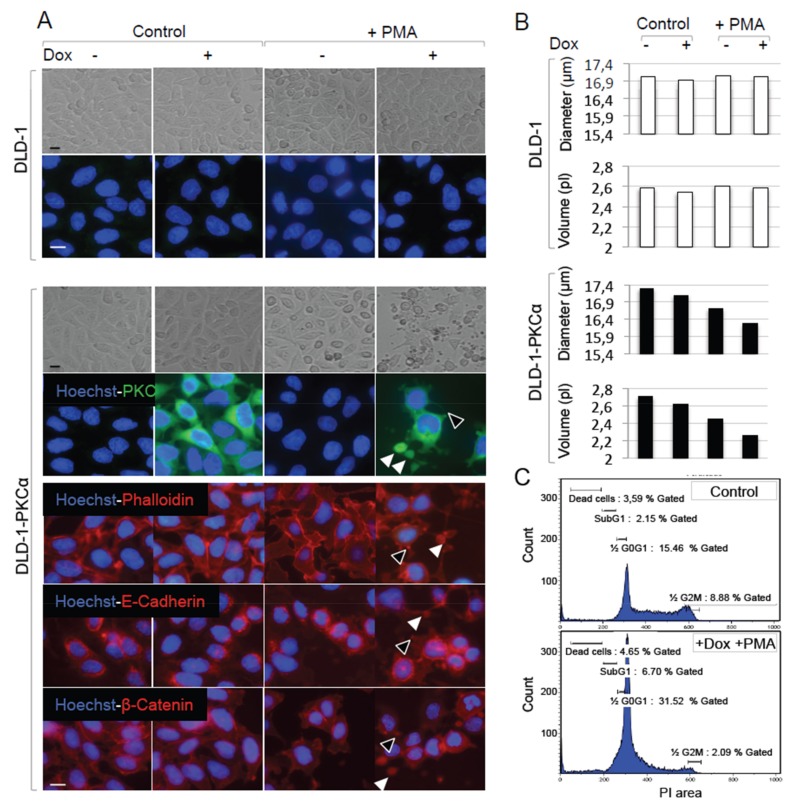
Inducing PKCα activity alters the shape, decreases the size, and stops cell cycling in DLD-1-PKCα CRC cells. (**A**) Parental and DLD-1 cells in which PKCα can be induced by doxycycline (DLD-1-PKCα cells) were incubated or not with 1 μM doxycycline (to induce PKCα expression) and/or 1 nM PMA (to induce PKC activity) for 24 h and were observed by phase contrast (first and third line) or by epifluorescence microscopy. Hoechst staining of the nuclei (blue) and staining for PKCα (green), actin (red; phalloidin), E-cadherin (red) and β-catenin (red), as indicated, show that increasing PKCα function in DLD-1-PKCα cells (doxycycline + PMA) alters cell morphology and induces a strong remodeling of the actin-E-cadherin-β-catenin network. Note the localization of overexpressed PKCα, actin, E-cadherin and β-catenin in membrane protrusions (empty arrowheads) and large buds (solid arrowheads) upon incubation with doxycycline and PMA (scale bar: 5 μm). (**B**) Scepter measurements carried out in DLD-1 control cells (upper panel) and DLD-1-PKCα cells (lower panel) exposed or not to 1 μM doxycycline and/or 1nM PMA for 24 h, show that increasing PKCα function in DLD-1-PKCα cells (doxycycline + PMA) significantly decreases the cell volume and diameter. (**C**) Flow cytometry analysis of DLD-1-PKCα cells exposed (lower panel) or not (Control, upper panel) to doxycycline + PMA for 24 h shows that increasing PKCα function in DLD-1-PKCα cells induces cell cycle arrest in the G0/G1 phase.

**Figure 6 cancers-11-00693-f006:**
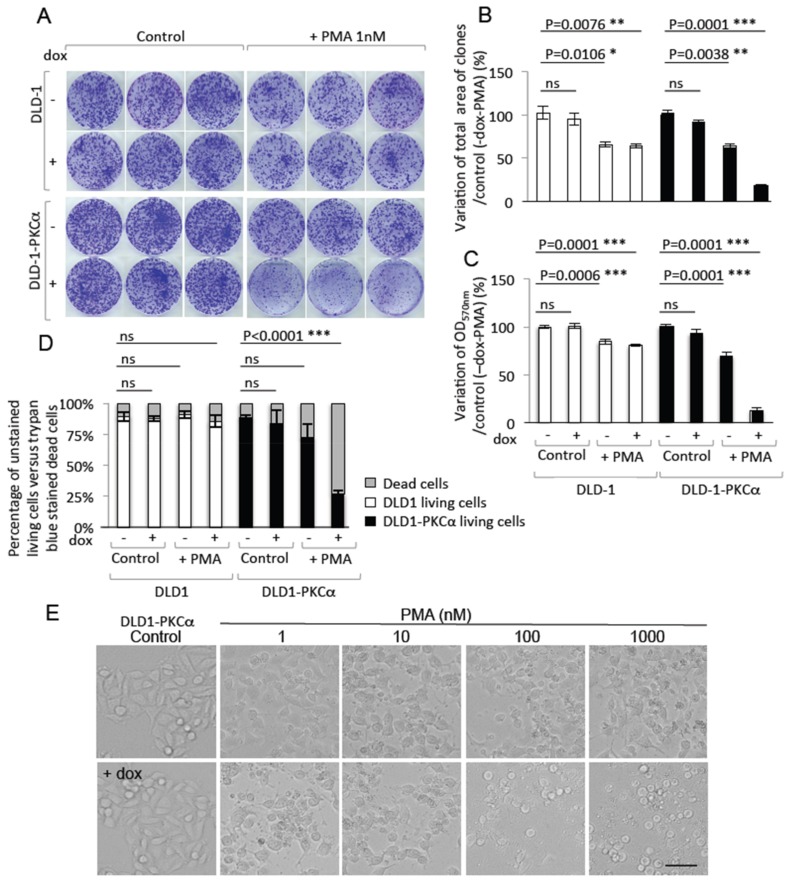
PKCα activity induces DLD-1 CRC cell death. (**A**) Crystal violet assays in DLD-1 control cells and DLD-1-PKCα cells incubated or not with 1 μM doxycycline and/or 1 nM PMA for 10 days. Quantification of the variations (**B**) of the total area occupied by the clones, and (**C**) of the OD_570 nm_ values (compared with the control condition without doxycycline and PMA) shows that increasing PKCα function in DLD-1-PKCα cells (doxycycline + PMA) leads to a reduction in clone formation due to cell growth inhibition. (**D**) Quantification of the percentage of trypan blue-positive DLD-1 and DLD-1-PKCα cells (i.e., dead cells; in grey) and –negative living cells (white: DLD-1 cells; black: DLD-1-PKCα cells) after exposure to 1 μM doxycycline and/or 1nM PMA for 10 days demonstrates that increasing PKCα function in DLD-1-PKCα cells (doxycycline + PMA) strongly induces cell death. (**E**) Phase-contrast images of DLD-1-PKCα cells incubated with increasing concentrations of PMA without or with 1 μM doxycycline for 24 h show that DLD-1-PKCα cell death is PMA dose-dependent. Scale bar: 20 μm. * *p* ≤ 0.05, ** *p* ≤ 0.01, *** *p* ≤ 0.001, ns: *p* > 0.05.

**Figure 7 cancers-11-00693-f007:**
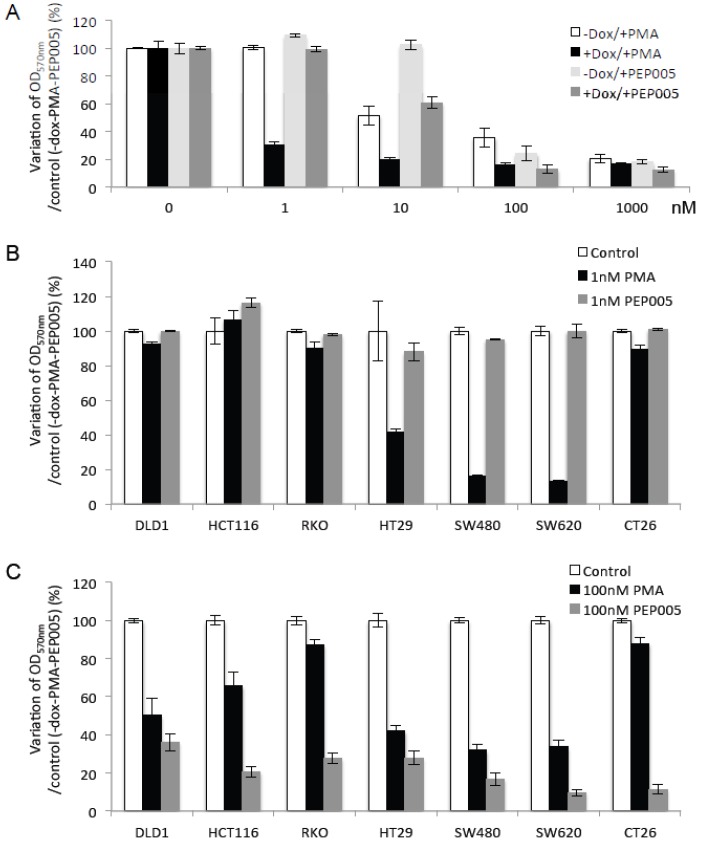
PMA and PEP005 di-terpene esters inhibit CRC cell growth in a concentration-dependent manner with distinct potentials. (**A**) Quantification of crystal violet assays in DLD-1-PKCα cells exposed or not to 1 μM doxycycline and increasing concentrations of PMA or PEP005, as indicated, for 10 days. OD_570 nm_ values (compared with the control condition in the absence of doxycycline and PMA or PEP005) show that increasing PKCα expression and activity by doxycycline and PMA or PEP005 strongly promotes growth arrest. Note that the required concentrations are 10 times higher for PEP005 than for PMA to reach similar effects. Crystal violet assay performed in a panel of CRC cell lines incubated or not for 10 days with (**B**) 1 nM PMA or PEP005 and (**C**) 100 nM PMA or PEP005. OD_570nm_ values (compared with the control condition without PMA or PEP005) show the strong effect of PMA at low concentration (1 nM) in a limited number of CRC cell lines. PEP005 did not have any effect at low concentration (1 nM), but showed strong growth inhibition at high concentration (100 nM) in all tested CRC cell lines.

**Table 1 cancers-11-00693-t001:** Clinical trials of natural PKC activators in human cancers (https://clinicaltrials.gov/).

Drug Names	Selectivity	Molecule	Targeted Cancer	Phase	Number of Studies					
Bryostatin 1	Classical and novel PKC	Macrolide lactone	Leukemia	**I, II**	9					
	2 fold for ε over α and δ	Naturally found in	Lymphoma	I, II	9					
		the Bryozoan species	Renal cancer	I, II	7					
		*Bugula neritina*	Unspecified adult tumors	I	6					
			Gastric cancer	II	3					
			Esophageal cancer	II	2					
			Melanoma (skin)	I	2					
			Myeloma	II	2					
			Ovarian cancer	II	2					
			Breast cancer	II	1					
			Cervical cancer	II	1					
			Colorectal cancer	II	1					
			Fallopian tube cancer	II	1					
			Head and neck	II	1					
			Lung cancer	II	1					
			Pancreatic cancer	II	1					
			Prostate cancer	II	1					
			Small intestine cancer	I	1					
12-O-tetradecanoylphorbol-13-acetate	Classical and novel PKC	Diterpene ester	Leukemia	I, II	2					
Phorbol 12-myristate 13-acetate	α, β,γ, δ, ε, θ, η	Naturally found in	Skin neoplasms	II	1					
TPA, PMA		the *Euphorbia*								
		*Croton Tiglium*								
Ingenol-3-angelate PEP005	Preferential for α and δ	Diterpene ester	Basal Cell Carcinoma	II	4					
		Naturally found in	Lentigo Maligna	II			1			
		the *Euphorbia Peplus*	Seborrheic Keratosis	II, III	6					
			Squamous Cell carcinoma	II	1					

completed 

; Currently recruiting participants 

.

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
