# Peer review of "Modulating PKCα Activity to Target Wnt/β-Catenin Signaling in Colon Cancer"

_cancers, 2019, doi:10.3390/cancers11050693_

Round 1

Reviewer 1 Report

The authors used both in vitro (DLD-1 cells) and in vivo (C57BL/6J mice) PKCα knock-in models to investigate whether enhancing PKCα function could be beneficial in colorectal cancer treatment.

They found that PKCα was infrequently mutated in CRC samples, and that inducing PKCα function was not deleterious for the normal intestinal epithelium. Conversely, di-terpene ester-induced PKCα activity triggered CRC cell death.

However, there were any evidence to show the connection between PKCα and Wnt/beta-catenin.

Therefore, more experiments should be conducted to show the link between PKCα and beta-catenin in CRC.

In addition, only one cell line was utilized in this study, and at least 2 difference cell lines were suggested to be used for drawing the conclusions.

Lastly, the figure quality was poor and should be improved.

Author Response

Review Report Form

Open Review -1

English language and style

( ) Extensive editing of English language and style required
( ) Moderate English changes required
(x) English language and style are fine/minor spell check required
( ) I don't feel qualified to judge about the English language and style

Yes

Can be improved

Must be improved

Not applicable

Does the introduction provide   sufficient background and include all relevant references?

(x)

( )

( )

( )

Is the research design   appropriate?

( )

(x)

( )

( )

Are the methods adequately   described?

( )

(x)

( )

( )

Are the results clearly presented?

( )

(x)

( )

( )

Are the conclusions supported by   the results?

( )

(x)

( )

( )

Comments and Suggestions for Authors

The authors used both in vitro (DLD-1 cells) and in vivo (C57BL/6J mice) PKCα knock-in models to investigate whether enhancing PKCα function could be beneficial in colorectal cancer treatment. They found that PKCα was infrequently mutated in CRC samples, and that inducing PKCα function was not deleterious for the normal intestinal epithelium. Conversely, di-terpene ester-induced PKCα activity triggered CRC cell death.

1- However, there were any evidence to show the connection between PKCα and Wnt/beta-catenin. Therefore, more experiments should be conducted to show the link between PKCα and beta-catenin in CRC.

2- In addition, only one cell line was utilized in this study, and at least 2 difference cell lines were suggested to be used for drawing the conclusions.

3- Lastly, the figure quality was poor and should be improved.

To Reviewer-1

The authors thank the Reviewer-1 for his interest in their work and for his wise comments to improve their manuscript. Please, find hereafter, our answers to these comments.

1-   Our explanations do not seem to be clear enough, and we apologize for that. Indeed, supplementary figure S2 demonstrates that inducing PKCα activity is associated with a decrease in Wnt/β-catenin activity in our genetically modified DLD-1-PKCα doxycycline-inducible model: we used the TOP-Flash reporter system, in which the promoter contains tandem TCF binding sites and the luciferase level indicates the activity of the Wnt/β-catenin pathway (Korinek et al., 1998, Mol. Cell. Biol.); we observed that the (doxycycline+PMA)-induced PKCα function in DLD-1-PKCα cells induces a decrease of luciferase activity (Figure S2A) and this, even in the presence of a stabilized β-catenin (figure S2B). In order to clarify our explanations, we have added the following comment together with the related reference at the end of the paragraph entitled “Increasing PKCα Activity in DLD-1 CRC Cells Disrupts Cell Morphology and Causes Cell Cycle Arrest” in the results section. “Since activating PKCα function and inhibiting the Wnt/β-catenin signaling both cause cell cycle arrest in the G1 phase (figure 4C and Tetsu et al. 1999, Nature), we can reasonably assume that PKCα-induced colon cancer cells growth arrest is mediated through inhibition of the Wnt/β-catenin signaling even if we can not exclude the involvement of additional mechanisms (Lines 169 to 173, Page 6). Furthermore, we included additional western blots in a new supplemental figure S3 that evidences a decrease of active β-catenin in doxycycline+PMA-treated PKCα-DLD-1 cells. The related comment has also been added in the text (Lines 168 to 169, Page 6).

2-   We agree with the reviewer that the DLD1-PKCα cell line was the only genetically modified colon cancer cell line model used in our experiments in order to artificially induce PKCα expression. This model was generated from the human DLD-1 colon cancer cell line since this parental cell line has the double advantage to display reduced PKCα expression level and a PKCβ LOF mutation which, at least, eliminates PKCβ activity (Antal et al. 2015, Cancer cell) and facilitates observation of the potential tumor suppressor effect of the doxycycline+PMA induced PKCα function. Thus, the purpose of creating this artificial model was to clearly establish whether or not inducing selectively PKCα function was relevant to induce a tumor suppressor activity in colon cancer cells. From a clinical point of view, we think that an additional artificial model generated from another parental colon cancer cell line would not necessarily be a significant added value. In addition, generating and analyzing phenotypic features of such an additional model can reasonably not be carried out before the deadline for sending revisions for the special issue on “targeting Wnt signaling in cancer”. Besides, we demonstrate the tumor suppressor effects of PKC activators on seven colon cancer cell lines (figure 7), which validate the artificial DLD1-based cell model. Finally, our aim is rather now to carry on the project by testing the anti-tumoral effects of vectorized PKC activators in vivo in xenografted mice.

3-   We agree with Reviewer-1 concerning the poor quality of the figures and we apologize for that. Figures were initially submitted in jpeg format and are now provided in pdf format which greatly improves their quality.

Reviewer 2 Report

In their manuscript entitled “Modulating PKCα Activity to Target Wnt/β-catenin Signaling in Colon Cancer,” Dupasquier et al. (1) analyze the frequency of inactivating mutations in PKCα in CRCs, determining that their incidence is low in patients (2.78-5.35%); (2) show that increased expression of PKCα in the murine intestine does not produce significant abnormalities; and (3) confirm many previous studies showing that activation of PKCα function in CRC with mutated APC inhibits cell growth.  The authors also determine that long-term treatment with di-terpene esters inhibits CRC cell growth.  One of the most interesting aspects of the study is the development of the first mouse model of PKCα overexpression in the intestine.  The analysis of mutations in colon cancer is also valuable.  Less compelling are the studies in cell lines, which are not novel.  The following concerns should be addressed:

Major Comments:

The authors have developed the first mouse model overexpressing PKCα in the intestine.  The finding that increased levels of PKCα do not significantly affect the intestinal mucosa are very interesting.  However, there is a missed opportunity to significantly advance the field by leveraging this valuable model.  The study would be strengthened if the authors treated their transgenic mice with e.g. a carcinogen that induces intestinal/colon tumor formation, such as azoxymethane (AOM) or a combination of AOM and DSS, and compared tumor development in the presence of increased levels of the kinase.

Fig. 4B: The authors should provide higher resolution images that clearly demonstrate the expression and subcellular distribution (e.g., membrane association) of PKCα in the intestinal epithelium.  This cannot be seen in the fluorescence images provided and the data are not informative as shown.

Fig. 4:  The authors should evaluate the expression of other members of the PKC family in PKCα-overexpressing tissue to determine if there are compensatory effects.

The finding that increased PKCα activity in CRC cells, including DLD1 cells, induces cell cycle arrest is not novel; this effect has been shown in many previous publications.  The authors need to reference these studies in the section starting on line 136: e.g., Ref. 12; Ref. 13; among others that were not included.

The authors should determine the effects of PKCα overexpression in DLD1 cells on β-catenin levels: are they decreased, as previously reported by Gwak et al?

Fig. 5A also requires higher resolution images to appreciate the data.

The experiments with di-terpenes are conducted over a period of many days. Can the effects be attributed to activation of PKCα?  Do these compounds affect the expression of the enzyme?  These questions need to be addressed.

 Minor Comments:

References should be provided for the following statement: PKCα activity is inversely correlated with that of the Wnt/β-catenin pathway: it gradually increases from the cryptic compartment to the surface of normal intestinal epithelium, while it is significantly reduced in CRC in which the Wnt/β-catenin signaling pathway is constitutively activated (Figure 1). 

In Fig. 4D, what are villosoties?

The authors state that PRKCA is mutated at a very high frequency in melanoma and skin cancer, based on data from http://dcc.icgc.org.  This is not supported by information in the TCGA.  It is important for the authors to clarify the discrepancy – at least mention it, accompanied with the numbers from the two sites – to provide a more balanced view of available information.

Author Response

Haut du formulaire

Review Report Form

Open Review

English language and style

( ) Extensive editing of English language and style required
(x) Moderate English changes required
( ) English language and style are fine/minor spell check required
( ) I don't feel qualified to judge about the English language and style

Yes

Can be improved

Must be improved

Not applicable

Does the introduction provide   sufficient background and include all relevant references?

( )

(x)

( )

( )

Is the research design   appropriate?

( )

( )

(x)

( )

Are the methods adequately   described?

(x)

( )

( )

( )

Are the results clearly presented?

( )

( )

(x)

( )

Are the conclusions supported by   the results?

( )

( )

(x)

( )

Comments and Suggestions for Authors

In their manuscript entitled “Modulating PKCα Activity to Target Wnt/β-catenin Signaling in Colon Cancer,” Dupasquier et al. (1) analyze the frequency of inactivating mutations in PKCα in CRCs, determining that their incidence is low in patients (2.78-5.35%); (2) show that increased expression of PKCα in the murine intestine does not produce significant abnormalities; and (3) confirm many previous studies showing that activation of PKCα function in CRC with mutated APC inhibits cell growth.  The authors also determine that long-term treatment with di-terpene esters inhibits CRC cell growth.  One of the most interesting aspects of the study is the development of the first mouse model of PKCα overexpression in the intestine.  The analysis of mutations in colon cancer is also valuable.  Less compelling are the studies in cell lines, which are not novel.  The following concerns should be addressed:

Major Comments:

1- The authors have developed the first mouse model overexpressing PKCα in the intestine.  The finding that increased levels of PKCα do not significantly affect the intestinal mucosa are very interesting.  However, there is a missed opportunity to significantly advance the field by leveraging this valuable model.  The study would be strengthened if the authors treated their transgenic mice with e.g. a carcinogen that induces intestinal/colon tumor formation, such as azoxymethane (AOM) or a combination of AOM and DSS, and compared tumor development in the presence of increased levels of the kinase.

2-Fig. 4B: The authors should provide higher resolution images that clearly demonstrate the expression and subcellular distribution (e.g., membrane association) of PKCα in the intestinal epithelium.  This cannot be seen in the fluorescence images provided and the data are not informative as shown.

3-Fig. 4:  The authors should evaluate the expression of other members of the PKC family in PKCα-overexpressing tissue to determine if there are compensatory effects.

4-The finding that increased PKCα activity in CRC cells, including DLD1 cells, induces cell cycle arrest is not novel; this effect has been shown in many previous publications.  The authors need to reference these studies in the section starting on line 136: e.g., Ref. 12; Ref. 13; among others that were not included.

5-The authors should determine the effects of PKCα overexpression in DLD1 cells on β-catenin levels: are they decreased, as previously reported by Gwak et al?

6-Fig. 5A also requires higher resolution images to appreciate the data.

7-The experiments with di-terpenes are conducted over a period of many days. Can the effects be attributed to activation of PKCα?  Do these compounds affect the expression of the enzyme?  These questions need to be addressed.

 Minor Comments:

8-References should be provided for the following statement: PKCα activity is inversely correlated with that of the Wnt/β-catenin pathway: it gradually increases from the cryptic compartment to the surface of normal intestinal epithelium, while it is significantly reduced in CRC in which the Wnt/β-catenin signaling pathway is constitutively activated (Figure 1). 

9-In Fig. 4D, what are villosoties?

10-The authors state that PRKCA is mutated at a very high frequency in melanoma and skin cancer, based on data from http://dcc.icgc.org.  This is not supported by information in the TCGA.  It is important for the authors to clarify the discrepancy – at least mention it, accompanied with the numbers from the two sites – to provide a more balanced view of available information.

To Reviewer-2

The authors thank the Reviewer-2 for his interest in their work and for his wise comments to improve their manuscript. Please, find hereafter, our answers to these comments.

Major Comments:

1-     We agree with Reviewer-2 that the study would be strengthened by evaluating the incidence of increased levels of PKCα on tumor development in our PKCα knock-in mice model treated with AOM or AOM+DSS. This is indeed the next step of our project. It has been shown that the molecular hallmarks of CRC development following AOM/DSS treatment are mainly due to an enhanced inflammatory immune response (reason why this model is also used in the context of studies on inflammatory bowel disease (IBD), Crohn’s disease (CD), or colitis-associated cancer or CAC). In addition, studies from this animal model have shown increased activity of the Wnt signaling pathway due to mutational activation of β-catenin. Sites of β-catenin mutations after AOM/DSS treatment have been described to be located in codons that play important roles in the degradation of β-catenin as described in Tanaka, 2012, Int  J Infl. Thus, it could be postulated that the use of AOM/DSS treatment may impair the anti-oncogenic effects of PKCα in our mice model. However, given our data that PKCα can inhibit the activity of the Wnt/β-catenin even upon over expression of a stabilized S33-β-catenin (see supplemental figure S2), we agree that it’s worth to be tested. Besides, we also started to cross our PKCα knock-in mice with the APCdelta14 mice in order to assess the effectiveness of increased PKCα expression in preventing the APCdelta14-induced tumor development in the intestine. However, taking into account the significant additional time required to cross the mice, collect the samples and analyze the resulting phenotypes, we plan to dedicate those data to a future article focusing on stimulating PKCα activity in pre-clinical models of colon cancers. Besides, the present study validates the PKCα knock-in model in a tumor-free background and demonstrates that increasing PKCα function in the intestine is safe, two critical criteria to pursue the study in pre-clinical models of colon cancers.

2-   We apologize for the poor quality of figure 4B. All Figures were initially submitted in jpeg format and are now provided in pdf format, which greatly improves their quality.

3-   From a fundamental point of view, it would be informative to assess whether there are compensatory effects of other PKC members. Unfortunately we do not have the appropriate antibodies in our laboratory right now and we apologize for not being able to provide those data before the upcoming deadline for sending revisions for the special issue on “Targeting Wnt Signaling in Cancer”. Besides, the result might not have significantly impact the message delivered in figure 4 i.e. that “Inducing PKCα Function is not Deleterious for the Normal Intestinal Epithelium “.

4-   We thank Reviewer-2 for this comment and included the following references in the article in addition to references 12 and 13  (Line 163, Page 6).

Pysz et al. Differential regulation of cyclin D1 expression by protein kinase C α and ϵ signaling in intestinal epithelial cells. J Biol Chem. 2014 Aug 8;289(32):22268-83. doi: 10.1074/jbc.M114.571554. Epub 2014 Jun 9.

Pysz et al. PKCalpha tumor suppression in the intestine is associated with transcriptional and translational inhibition of cyclin D1.Exp Cell Res. 2009 May 1;315(8):1415-28. doi: 10.1016/j.yexcr.2009.02.002. Epub 2009 Feb 14.

Guan et al. Protein kinase C-mediated down-regulation of cyclin D1 involves activation of the translational repressor 4E-BP1 via a phosphoinositide 3-kinase/Akt-independent, protein phosphatase 2A-dependent mechanism in intestinal epithelial cells. J Biol Chem. 2007 May 11;282(19):14213-25. Epub 2007 Mar 13.

Hizli et al. Protein kinase C alpha signaling inhibits cyclin D1 translation in intestinal epithelial cells. J Biol Chem. 2006 May 26;281(21):14596-603. Epub 2006 Mar 23.

These studies indeed demonstrate a PKCα-mediated control of both transcriptional and translational regulation of Cyclin D1 which is itself a gene target of the Wnt/β-catenin signaling. In addition, this PKCα-mediated control is associated with cell growth inhibition for several colon cancer cells including DLD-1, HCT-15, Colo205, SW620. Besides, PMA concentrations used in those studies are usually ranging from 10 to 100nM, while we show a differential sensitivity of colon cancer cell lines at low PMA concentrations which seems to discriminate colon cancer cells with a MSI status from those with a MSS status (Lines 208 to 210, Page 7) : “All PMA-sensitive CRC cell lines (HT29, SW480, SW620) displayed a Microsatellite Stable status (MSS), while the others (DLD-1, HCT116, RKO and CT26) exhibited a Microsatellite Instable status (MSI).”

5-   We thank Reviewer-2 for this comment and we provide additional western blots in a new supplemental figure S3 that evidences a decrease of active β-catenin in doxycycline+PMA treated PKCα-DLD-1 cells. The related comment has also been added in the text (Lines 168 to 169, Page 6).

6-   As for figure 4B, we apologize for the poor quality of figure 5A and as explained above (answer 2-) all figures are now provided in pdf format, which greatly improves their quality.

7-   Addressing the question as to whether the effects observed in response to di-terpene exposure over a long period of time can be attributed to activation of PKCα was challenging and that’s one of the reasons we generated our artificial PKCα-DLD-1 model. Indeed, this model ensures a constant production of PKCα in DLD-1 cells in response to doxycycline exposure and it is precisely under these conditions that di-terpenes esters cause massive cell death, thus demonstrating that a sustained PKCα activity is required to induce these effects. This question is more difficult to address in non- genetically modified colon cancers cells. Indeed, even if a constant amount of PKCα is available for activation by di-terpene esters (i.e. a sustained balance between expression and di-terpene esters induced-down regulation of PKCα), the cell death occurring in the same time associated with the low levels of PKCα detectable in colon cancer cells makes the study technically difficult to perform.

 Minor Comments:

8-   References are now provided for studies reporting the gradual increase in PKCα activity from the cryptic compartment up to the surface of the normal epithelium while it is significantly reduced in CRC (Line 80, Page 2).

Frey et al. Protein kinase C isozyme-mediated cell cycle arrest involves induction of p21(waf1/cip1) and p27(kip1) and hypophosphorylation of the retinoblastoma protein in intestinal epithelial cells. J Biol Chem. 1997 Apr 4;272(14):9424-35.

Saxon et al. Activation of protein kinase C isozymes is associated with post-mitotic events in intestinal epithelial cells in situ. J Cell Biol. 1994 Aug;126(3):747-63.

Dupasquier et al.  A new mechanism of SOX9 action to regulate PKCalpha expression in the intestine epithelium. J Cell Sci. 2009 Jul 1;122(Pt 13):2191-6. doi: 10.1242/jcs.036483. Epub 2009 Jun 9.

Assert et al. Anti-proliferative activity of protein kinase C in apical compartments of human colonic crypts: evidence for a less activated protein kinase C in small adenomas. Int J Cancer. 1999 Jan 5;80(1):47-53.

Klein et al. Adenoma-specific alterations of protein kinase C isozyme expression in Apc(MIN) mice. Cancer Res. 2000 Apr 15;60(8):2077-80.

9-   Villosoties has been corrected into villi in figure 4D.

10- We thank Reviewer-2 for his comment about the discrepancy between ICGC and TCGA websites about the percentage of PKCα mutations found in skin cancers. We have now included the following comment in the manuscript that provides an explanation for these results:PRKCA mutations can be found in many human tumors, sometimes with high frequencies (e.g., 90.71% in skin cancer and 88.00% in melanoma) (MELA-AU, SKCA-BR, http://dcc.icgc.org). However, these observations must be tempered by the results displayed in the TCGA database (TCGA-SKCM, https://portal.gdc.cancer.gov) that identify PRKCA mutations in only 4.48% of skin cancers. This discrepency can be explained by the use of distinct sequencing strategies i.e. Whole Genome Sequencing (WGS) for ICGC and Whole Exome Sequencing (WXS) for TGCA. (Line 239, Page 12) .  In other words, the ICGC website lists PRKCA mutations found in both intronic and exonic sequences of the gene encoding for PKCα while the TCGA only lists PRKCA mutations found in exonic sequences, which explains the much lower number of mutations identified in the TCGA website.

Bas du formulaire

Round 2

Reviewer 1 Report

The authors have addressed almost all concerns.

Author Response

We thank Reviewer-1 for his interest in our work and for his concern about further improving the quality of our manuscript. As suggested, we have performed additional corrections concerning English language and style.

Best Regards,

Dr Corinne Prévostel

Reviewer 2 Report

This study should be of interest to investigators in the fields of PKC signaling and colon cancer.  The authors have responded to many of the concerns raised in the initial review.  However, some remaining issues need to be addressed:

1)      The resolution in Fig 4B is still too low to allow clear appreciation of the data.  Higher resolution images may also help clarify why levels of PKCalpha appear higher in the crypt than in the villus region (if this is indeed the case).

2)      Can the authors comment on the finding that PKCalpha transgenic mice exhibit increased numbers of crypt/villus units and colonic crypts? What are possible explanations for this result?

3)      Fig. 4D still refers to villosities. 

4)      In Fig. 4D, it is not clear what the authors mean by number/2 µm (an apsorptive intestinal epithelial cell is about 20 µm in height and microvilli in the brush border are 1-2 µm in height).

5)      Since the authors cannot provide conclusive support for the involvement of PKCalpha in the results they show in DLD1 cells, they should, at a minimum, provide evidence that levels of PKCalpha protein are maintained in Dox-treated cells at different times during the 10-day treatment with PMA or PEP005.

6)      Can the authors explain the absence of beta-catenin degradation in response to PKCalpha activation as previously reported in DLD1 cells and other cell types? 

7)      In my version of the manuscript, the references are ‘double numbered.’

8)      The authors conclude that “PKCα integrity is preserved in most human CRCs and can thus be functionally activated.”  However, many CRCs do not express the enzyme and presumably would not respond to therapies targeting PKCalpha.  This caveat should be incorporated into the discussion.

9)    a control experiment is still missing

Author Response

We thank Reviewer-2 for his interest in our work and for his concern about further improving the quality of our manuscript.

It seems that the reviewer could not see the new version of our figures (already taking into account points 1) and 3)) that have been provided in a separate pdf file during the last re-submission. Indeed, corrected figures have not been included in the last word document formatted by “Cancers” instead of “former” figures but have been provided in a separate pdf file in order to allow “Cancers editors” to reformat the article with better-quality images.

To allow the reviewer to see our corrections/improvements in figures, we now provide corrected figures as inserts in the word document formatted by “Cancers”.

Besides, please find below our answers to the reviewer’s comments and suggestions:

1)  As mentioned above, the resolution of figure 4B had been improved during the first round of review, and we hope that the Reviewer will now be able to see the changes made to figure 4B. We agree with the Reviewer that PKCα expression seems visually higher in the crypts of the small intestine epithelium, although we believe that this is not the truth because:

-    The PKCα expression levels can hardly be compared between the crypt and the villi given the different subcellular distribution of PKCα in the two compartments, i.e. cytoplasmic in cryptic cells and partly concentrated at the plasma membrane in the villi.

-    PKCα expression is no higher in the crypts than on the surface of the colon epithelium.

2)  The increase of the villi size observed in the mice knocked-in for PKCα could be due to an increase of cell differentiation resulting in accumulated differentiated cells along the villi. Besides, the increase in the number of villi/crypts is only an observation related to our knock-in model. To our knowledge such a phenotype has not been reported for other knock in/out models, and we can hardly speculate on the meaning of this observation. One could however postulate that the increase of PKCα activity may possibly interfere with the intestinal epithelium morphogenesis. This comment has now been added in the paragraph entitled “Increasing PKCα Activity in the Healthy Intestinal Epithelium is not Deleterious ” (lines 144-145).

3)  As explained above, “villosities” has already been replaced by “villi” in the new version of figure 4D during the first round of review, and we hope that the reviewer will now be able to see the corrections made to figure 4D, including those concerning the mistake about the measurement unit mentioned in point 4).

4)  We thank the reviewer for this comment because there was indeed a mistake for the measurement unit, which is, of course, the number of crypt/villi counted per 2mm (and not 2μm) tissue sections. This point has now been corrected.

5)  As explained in our previous answer to Reviewer-2, the DLD-1-PKCα model has precisely been generated in order to ensure a doxycycline-induced sustained production of PKCα in DLD-1 cells. This is obviously the case since it is only upon doxycycline exposure that di-terpene esters cause a massive cell death. As noticed by the Reviewer we did not provide evidence that PKCα expression is maintained at day 10 upon doxycycline+di-terpene ester exposure. In fact, we intended to do so but this issue was technically difficult to address given that 80 to nearly 100% of the doxycycline+di-terpene ester treated DLD-1-PKCα cells are dead at day 10. The best we could do to address this issue was to treat our cells no more than for 4 days in order to harvest enough cells to perform Western blots. The results, presented in the supplemental figure S3, clearly show a sustained amount of PKCα after 4 days exposure to doxycycline either in the absence (lane 3) or the presence of PMA (lane 4). Besides, maybe our choice to expose our DLD-1-PKCα cells to both doxycycline and di-terpene ester over such a long period of time is surprising for the reviewer. In fact, the experiment has been designed as such in order to check whether or not doxycycline+di-terpene ester treatment was able to induce a sustained inhibition of cell growth over a long period of time. Since the answer is yes, we consider that it is now relevant to test this effect on tumor growth in vivo, in grafted mice models.

6)  As shown by the Western blots presented in the supplemental figure S3, we did not see a significant decrease in total β-catenin in doxycycline+PMA treated PKCα-DLD1-cells even if the amount of total β-catenin had a tendency to decrease (lane 4 compared with lanes 1, 2, 3). Besides, and more interestingly, we clearly observed a decrease in the amount of active β-catenin in those conditions. As noticed by the Reviewer, we did not observe obvious similar effects in the di-terpene treated parental cell line. This is possibly due to the low levels of endogeneous PKCα in DLD-1 cells but also to the low concentration of di-terpene esters (i.e. 1nM) used in our experiments. To our Knowledge, Gwack et al. (2006, J. Cell Sci.) did not induced PKCα activity in order to demonstrate that Protein-kinase-C-mediated β-catenin phosphorylation negatively regulates the Wnt/ β-catenin pathway but rather inhibited PKC activity with BIM in HEK293 reporter cells. Few years later, they however reported a decrease in β-catenin amounts in SW480 and DLD-1 cells treated with  PMA but when using 10 and 50nM concentrations (i.e concentrations 10 and 50 fold higher than the one used in our study) (Gwack et al. (2009, Mol. Targets). Lee et al. (2010, Cell) also demonstrated that RORα attenuates Wnt/β-Catenin signaling by PKCα-dependent phosphorylation in colon cancer, but again, used reporter strategies in HEK293 and HCT116 cells, thus suggesting that di-terpene ester-induced significant effects on the amount of total β-catenin in non genetically modified cell line may possibly be difficult to clearly evidence.

7)  The doubled number before each reference has been deleted.

8)  We agree with the Reviewer that PKCα expression levels are decreased in many CRC cells including in tumors spontaneously developing in APC min mice (Klein et . 2000, Cancer Res; Oster and Leitges, 2006, Cancer Res.). As requested by the reviewer, a caveat has been included in the discussion section (lines 317-325). However, this does not mean that PKCα is fully knocked down and this is consistent with our observation that all CRC cell lines tested stop growing upon di-terpene ester treatments (figure 6), (even if we can not exclude activation of other members of the PKC family). In addition, Gwack et al (2009) (cited above in point 6)) could detect low levels of PKCα in both SW480 and DLD-1 cells by western blot. Besides, Assert et al. (1999, Int J. cancer reported a significant increase of PKCα in the cytosolic fraction of colonic adenomas compared to normal, neighboring mucosa, thus suggesting, again, that even if PKCα expression is decreased in CRC cells, it is still expressed but not activated. Interestingly, this observation raises the question of whether the PKCα expression levels detected in colon adenomas are evolving or not with tumor progression. Addressing this issue could be critical to clarify whether targeting PKCα would be more relevant at early, late, or any stage of the disease. Thus, a comment has been added in the discussion regarding this issue (lines 327-331).

9)  We apologize, but, regarding this comment, we do not understand what control the Reviewer is referring to.

Best Regards,

Dr Corinne Prévostel

Round 3

Reviewer 2 Report

The authors have revised accordingly, all is fine.

Author Response

All requested minor revisions performed in the manuscript have been highlighted using the "Track  Changes" function in Microsoft Word) : 

1.     line 199 : For the avoidance of over-interpretation, ‘we can reasonably assume that PKCα-induced colon cancer cells growth  arrest is mediated through inhibition of the Wnt/β-catenin signaling even if  we can not exclude the involvement of additional mechanisms’ as been changed to ‘we conclude that PKCα-induced colon cancer cell growth arrest is mediated  through inhibition of Wnt/β-catenin signaling. However, we cannot exclude  the involvement of additional mechanisms’.

2.     line 396 : ‘Besides’ has been deleted.

3.      All numbers and units have been separated by a space. 

4.     Grammar errors have been corrected:

-   Line 297 (legend of figure 6): Figure ‘induces DLD-1 CRC cells death, improve drugs efficacy has been changed to ‘induces DLD-1 CRC  cell death'

-   Line 406: 'drugs  efficacy’ has been changed to drug efficacy.

5.     Line 358: ‘differently from what reported for the epidermis’ has been changed to ‘differently from what was reported for the epidermis.

6.     Lines 333, 362: relevant ‘figure 2’ and  ‘figure 5, lower panel’ have been stated.

7.      ‘(A representative experiment on the two experiments carried out)’ has been added at the end of the legends of supplementary Figures S2 and S3 in order to indicate that the experiments were done two times.

8.     ‘PKCa’ has been changed to ‘PKCα’ when required.

9.     ‘R26-PKCαTg/Tg’ has been changed toR26-PKCαTg/Tg’ when required.

We hope that our revisions are satisfactory for proceeding further,

Best regards,

Dr Corinne Prévostel